# The Impact of Time Pressure on the Results of Psychotechnical Tests Based on the Findings of Pilot Studies Conducted on a Group of Students of the Silesian University of Technology—A Case Study

**DOI:** 10.3390/ijerph192214724

**Published:** 2022-11-09

**Authors:** Zygmunt Korban, Maja Taraszkiewicz-Łyda

**Affiliations:** Department of Safety Engineering, Faculty of Mining, Safety Engineering and Industrial Automation, Silesian University of Technology, 44-100 Gliwice, Poland

**Keywords:** stress, time pressure, psychotechnical tests, the analytic hierarchy process method

## Abstract

The performance of job tasks is increasingly being viewed through the prism of time constraints. Stress, as a consequence of these constraints, can play a dual role: motivating or destructive. This paper addresses the role of time pressure during the implementation of psychotechnical tests. Based on the example of a pilot group, which consisted of students of the Faculty of Mining, Safety Engineering, and Industrial Automation of the Silesian University of Technology, the authors presented the results of the study of the impact of time regime on the assessment of visuomotor coordination, psychomotor reaction time, the ability to focus attention, perceptual speed and accuracy, attention divisibility, and set-shifting. With the use of a survey conducted among the subjects and multivariate analysis (the Analytic Hierarchy Process method), an assessment was made in terms of difficulty levels during the implementation of exercises, including four evaluation criteria: complexity, repetition and timing of emitted signals, and the required accuracy of response to the signals in question. In the process of verifying the consistency of the evaluations carried out, the consistency ratio (CR) was adopted.

## 1. Introduction

Nowadays, time pressure in the work environment is almost universal. It is most often listed among other stressors for employees. Time-pressured work should be understood as a type of work in which the employee feels the burden of short deadlines for the completion of tasks, which often translates into the occurrence of stress [1]. In addition to the mentioned factor (time pressure), there may be other causes of stress (both in daily life and at work), such as environmental conditions, living conditions, workplace environment, work process, relationships with people, habits, and daily activities.

However, there is a group of activities where mainly time pressure has the greatest impact on the occurrence of stress, which translates into the quality of work performed. Moreover, this factor can be verified on the basis of specific and required psycho-technical tests. That is why, in accordance with current legislation, there is a need for such tests to assess the psychophysical fitness and ability of individuals to perform specific work.

Accordingly, there is a need for psychotechnical tests involving a specific group of workers. The authors measured visuomotor coordination, psychomotor reaction time, the ability to focus attention, perceptual speed and accuracy, attention divisibility, and set-shifting among the students of the University of Technology, namely future potential employees forced to face time pressure and the associated stress. This undertaking was aimed at raising the competence and awareness of students in the area of work ergonomics, including safety, hygiene, and physiology in their broadest sense.

The information relations between the object and the human in the control process, especially the perception of signals and the reactions of the worker (reaction time, the manner of response, and the adequacy of response to the received signals) are the basis of occupational safety at the workstation. Human characteristics and biological effects of work are an important part of the diagnosis, both in the field of work ergonomics and OSH. That is why it was so important to initially conduct the study and compile the results.

## 2. Theoretical Basis of the Study

### 2.1. Long-Term Stress, Causes, and Symptoms—Literature Review

Stress involves bidirectional communication between the brain and the cardiovascular, immune, and other systems via neural and hormonal mechanisms [2].

H. Selye was the first to use the concept of stress in medicine in 1926, defining the term as a non-specific reaction of the body to any external requirement manifested as psychological as well as physiological reactions [3,4].

Stress can be defined as a subjective and cognitive appraisal of a situation that taxes or exceeds an individual’s resources and that is experienced as overwhelming. It can result in an interplay of psychological (e.g., emotional arousal and valence) and physiological responses (e.g., variations in heartbeat, sweat production, skin temperature, and voice) which arise within seconds and may last from minutes to hours [5].

Stress is also defined as a state of mental tension caused by discrepancies between the demands of the environment and a person’s capabilities. These capabilities make an individual perceive these discrepancies as a threat to their health, life, or integrity. Factors that threaten these values are called stressors [6].

The factors (stressors) that cause stress at work can come from a variety of sources. They may be triggered by work itself, but also by the conditions under which it is performed, including its technology. Scientists regard stressors as an element of a stimulus or situation that disturbs the relative balance between the subject and the environment, and consequently, activates the adaptive mechanisms of human behavior [7].

Prolonged stress in the workplace can cause physical and psychological exhaustion, leading to somatic complaints. When such situations are repeated multiple times, they can, over time, lead to health issues, such as muscle pain in the neck, shoulders, and sacral region, ulcerations of the digestive system, gastrointestinal disorders, a weakened immune system and related infectious diseases, and an increased risk of cancer. Additionally, chronic psychological stress is associated with a higher risk of depression, cardiovascular disease (CVD), diabetes, autoimmune diseases, upper respiratory infections (URI), and poorer wound healing. Additionally, psychological stress is known to be associated with increased atherosclerosis, hypertension, and coronary heart disease (CHD) [8,9,10,11,12]. Stress has also been shown to affect inflammatory markers, cardiovascular risk, and other conditions, such as psoriasis and rheumatoid arthritis [13]. Based on different studies, it has been shown that a stressful work environment can be a factor in risky health behaviors, thus highlighting the importance of identifying groups of people at risk for drug abuse. Another study looked at the correlation between work-related stress and the use of prescription drugs for cognitive enhancement (CE), other stimulants (including cigarette smoking), and sleep disturbance. Thus, it has been found that alcohol, tobacco, or drug abuse can cause other conditions [14,15]. Work-related stress has been documented to be associated with the development of musculoskeletal disorders (WMSD) [16].

It also relates to economic impact. Individuals are unable to work as effectively and efficiently as before, make unintentional mistakes and errors, are reluctant to change or to engage with new things at work, tend to bypass regulations, safety rules, supervisors’ orders, take sick leave or days off, get injured at work, lose interest in work, and, in extreme cases, can resign and quit [17]. In this way, chronic stress in the workplace can increase the cost of business operations and lead to losses. Situations such as erroneous decisions made in haste, a decrease in efficiency and productivity at work, dissatisfied customers, the cost of workplace accidents, replacements for absent employees, selection, recruitment, and training and professional preparation of new employees or managers hired to replace those who have left are just examples of the costs that can be caused by unnecessary stress in the workplace [18]. Knowledge of the impact of the psychosocial work environment on the occurrence of stress-related disorders (SRDs) can help occupational physicians assess the relationship between these disorders and work [19,20,21].

### 2.2. The Essence of Psychotechnical Testing, Its Scope and Role

The aim of the study was to assess how students of the Faculty of Mining, Safety Engineering, and Industrial Automation at the Silesian University of Technology cope with performing tasks/activities under time constraints. These individuals can become managers and executives, who will be expected, among other things, to have the right way of perceiving the environment (including divisibility of attention, ability to select information, etc.) or the ability to solve complex problem situations often under time deficit conditions. In a situation where the role of a manager/supervisor concerns mainly supervision/observation of the implemented production process and correction of its course, certain psychophysical predispositions are of particular importance. The presented research results are a prelude to the planned research on the target group (representative group).

According to Section 6 of Article 210 of the Labor Code (“Rights and Duties of an Employee”), the Minister of Labor and Social Policy, in consultation with the Minister of Health and Welfare, determines the types of work that require so-called “special psychophysical fitness”. This fitness is verified by psychotechnical examinations, which assess mental fitness, as well as the ability of individuals to perform specific work related to the operation of machinery and equipment [22]. For a detailed list of work for which special psychophysical skills are required, see the bibliographic items [22,23,24,25,26].

In addition, according to the guidelines of the Provincial Center for Occupational Medicine and the National Consultant for Occupational Medicine, psychotechnical examinations are required to be performed by all employees hired as drivers, i.e., suppliers, couriers, drivers of emergency vehicles, as well as self-employed persons performing road transport activities.

Psychotechnical examinations include observation, a direct interview with a psychologist and examinations carried out using diagnostic tools (written psychotechnical tests, instrument tests). The necessity of psychotechnical examinations is determined by an occupational physician when conducting preventive examinations, to which the employer refers the employee (a referral is not required in cases of people who finance the examinations themselves). The referral should contain detailed information about the position that the employee holds/will hold, the work that the employee will perform along with information about the dangers present at the workplace. In the event of a change of employment, the employer and the occupational physician may or may not honor the results of tests performed earlier.

In the event of a negative psychotechnical test result:Examiners, instructors, cab drivers, professional drivers and drivers of emergency vehicles may, within 14 days of receiving the ruling, file an appeal (the appeal is filed through the psychologist who issued the ruling to the Provincial Center for Occupational Medicine, where reexamination will be conducted);Miners (or the entity referring them for examination), within 7 days of receiving the ruling, may file an appeal against the decision (the appeal is filed through the psychologist who issued the ruling to the Provincial Occupational Medicine Center, where reexamination will be conducted).

Additionally, in the case of drivers referred for examination by the mayor of a city, the district governor or the police, the result of the ruling is not a final decision: the person examined (or the entity referring them for examination), within 14 days of receiving the ruling, may appeal to its content (through the psychologist, to the Regional Occupational Medicine Center).

In the case of people working at heights, company car drivers, and machine operators, there is no appeal procedure. In these cases, the doctor decides on further proceedings.

Within the framework of the article, the authors proposed to assess the predisposition of technical university students to cope with the performance of tasks under time constraints (in the first stage only as a pilot study). This is due to the fact that in the future, these people may be subjected (in accordance with the regulations in force in Poland) to a similar type of tests to assess their psychophysical fitness and ability to perform specific jobs. Moreover, the analysis of the literature indicates that studies of a similar type have not been conducted (at least studies completed with publications), which, in the opinion of the authors, allows for the conclusion that the results contained in the article are unique, and further continuation of the research is fully justified.

## 3. Material and Methods

### 3.1. Instrument Tests

Within the psychotechnical tests conducted, instruments and tests in which reaction time plays a key role were used. Thus, the studies were to:Assess visuomotor coordination, psychomotor reaction time, the ability to focus attention, and perceptual speed and accuracy (Piórkowski apparatus, cross apparatus);Assess the speed and uniformity of responses to light and sound stimuli (reaction time meter);Assess coordination, attention divisibility, and set-shifting (Poppelreuter tables).

Due to the pilot nature of the study, and therefore, the limited scope of its conduct, the authors of the article consciously decided not to use a vortex flow meter (evaluation of rotational speed), a cubicle darkroom equipped with the Landolt ring or a noctometer (testing of twilight vision and evaluation of sensitivity to glare), the Raven’s Matrices test (progressive matrices test), or the Benett test, while keeping in mind their use at the stage of research conducted on the entire population of students of the Faculty of Mining, Safety Engineering, and Industrial Automation of the Silesian University of Technology.

While processing “raw” results, the sten scores [27] were used:sten score = 5.5 + 2 z_i_,(1)where z_i_ is the “raw” score after standardization:(2)$$z_{i}=\frac{x_{i}-\bar{x}}{s},$$
where is the x_i_ is the “raw” score and s is the standard deviation.

On this scale, the gradation is as follows:Sten scores of 1–4 (low level);Sten scores of 5–6 (average level);Sten scores of 7–10 (high level).

### 3.2. Analytic Hierarchy Process (AHP) Method, Theoretical Basis, and Applicability in the Process of Preference Research

In the broader diagnostic processes, both differentiation and determination of similarities between “objects” play an important role. In this type of activity, both quantitative and qualitative information is used (in both cases, a finite set of evaluation criteria is taken into account), and the very process of differentiating/defining similarities of “objects” is seen as a process related to the search for solutions within the framework of multi-criteria tasks. When dealing with the qualitative nature of the adopted evaluation criteria, one of the commonly used methods that allow the development of comparative statements is the Analytic Hierarchy Process (AHP) method [28,29,30,31,32,33,34,35,36,37,38,39,40,41,42,43,44]. The AHP method is counted among the set of methods used in the process of solving multi-criteria decision-making problems. In this set, the following can be distinguished:Methods using reference points, in which objects (variants) are compared with abstract reference solutions. Examples of these methods include Technique for Order Preference by Similarity to Ideal Solution (TOPSIS) and VIsekrzterijumskaOptimizacija i KompromisnoResenje (VIKOR);Additive methods, in which a matrix of normalized evaluations is determined, and an object (variant) is selected, for which the sum of evaluations is the highest. Examples of methods within this group are the simple additive weighting method (SAW) and fuzzy simple additive weighting method (F-SAW);Verbal methods based mainly on qualitative parameters, for which no objective aggregation model can be developed. The ZAPROS method (III) is an example of a method that belongs to this group;Analytical hierarchy methods and related methods. In this group of methods, independent criteria and objects (variants) are compared with each other in pairs, which makes it possible to create a scale vector and order the objects (variants). This group includes the analytical hierarchy process methods;Methods of the ELECTRE family, in which objects (variants) are evaluated according to maximized criteria and the final result is a superiority relation. The final result of the method is a graph of relationships between objects;PROMETHEE methods. In this group of methods, objects are compared in pairs due to the adopted evaluation criteria; for each pair of objects (variants), the so-called preference flows are determined.

In the Analytic Hierarchy Process (AHP) method, which is included in the group of so-called discrete multi-criteria decision support methods, the indicator of the relative importance of the criterion factor K_i_ over K_j_ (parameter a_ij_)—which is the quotient of the above-mentioned indicators—is determined:a_ij_ = r_i_/r_j_,(3)
where r_i_ is the absolute rank (significance) of the criterion K_i_ for i = 1, 2, …, n and r_j_ is the absolute rank (significance) of the criterion K_j_ for j = 1, 2, …, k.

Each comparison is assigned a verbal rating and a numerical value: “object” x_i_ compared to “object” y_i_ with respect to the criterion under consideration may be as follows:Extreme (numerical rating takes value 9);From very strong to extreme (8), very strong (7);From strong to very strong (6);Strong (5);Moderate to strong (4);Moderate (3);Equivalent to moderate (2);Equivalent (1).

The coefficients a_ij_ (a_ji_ = 1/a_ij_ for i = 1, 2, …, n) are numbers expressing the preference of the evaluator and are further components of the square matrix. Due to the adopted method of evaluation, this is the so-called proportional matrix, the properties of which are useful in determining the relative importance (weights) and in examining the degree of consistency of the compared elements.

Proceedings of the analytic hierarchy process method include the following steps [28,29,30,31,32,33,34,35,36,37,38,39,40,41,42,43,44]:Selection of n (i = 1, 2, …, n) “objects” and k (j = 1, 2, …, k) criteria to be evaluated;Pairwise comparison of the importance of “objects” and decision criteria;Development of a matrix of comparisons due to jth criteria (partial rankings);Development of the final ranking of solutions due to the adopted criteria;Determination of relative weights for k criteria;Determination of relative weights for the final ranking of criteria—selection of “object”.

In the process of verifying the consistency of the comparisons (evaluations) carried out, two measures are commonly used [28,29,30,31,32,33,34,35,36,37,38,39,40,41,42,43,44]:Consistency index (CI):
(4)$$\mathrm{CI}=\frac{\lambda_{\max}-n}{n-1},$$
Consistency ratio (CR):
(5)$$\mathrm{CR}=\frac{\mathrm{CI}}{r}=\frac{\lambda_{\max}-n}{r\;\left(n-1\right)},$$where λ_max_ is thethe maximum eigenvalue of the matrix, n is the number of objects assessed, and r is the random index.

Ratings are assumed to be consistent if the CR is no greater than 0.1.

## 4. Results and Research Findings—A Case Study

### 4.1. Results of Psychotechnical Tests among the Students of the Silesian University of Technology

The subjects were students of technical faculties at the University of Technology. The study group was selected due to the nature of their studies (technical studies, engineering) and the nature of their future work. The subjects were aged 19–25 years.

The content of this paper refers to the results of research in which the time regime plays a key role (Piórkowski apparatus, cross apparatus, reaction time meter, Poppelreuter tables). The research was conducted between April and July 2022 and had a pilot character (the so-called research reconnaissance). The study involved a group of 78 participants (36 women and 42 men) who were full-time students (aged 19 to 25) in three of the five currently active majors at the Faculty of Mining, Safety Engineering, and Industrial Automation of the Silesian University of Technology, that is:Geodesy and Cartography (21 subjects);Mining and Geology (8 subjects);Safety Engineering (49 subjects).

Time is the common element of tests to assess visuomotor coordination, psychomotor reaction time, the ability to focus attention and perceptual speed and accuracy (Piórkowski apparatus, cross apparatus), the speed and uniformity of reaction to light and sound stimuli (reaction time meter), as well as coordination, attention divisibility, and set-shifting (Poppelreuter tables): in the case of the Piórkowski apparatus, cross apparatus, and meter—the reaction time to the emitted signals—and in the case of Poppelreuter tables—the acceptable time to complete the task. The test results are presented in Figure 1.

It was only the distribution of the results (a sten scale was used), obtained based on the Poppelreuter test, that showed significant differences from the normal distribution. In the case of Poppelreuter tables, the share of low (1–4) sten scores and average (5–6) sten scores obtained was found to be comparable, namely 25.6% and 24.4%, respectively. The share of high scores was reported to be as high as 50%. This indicates the respondents’ ability to focus their attention on the problem being solved, as well as the divisibility of their attention (as part of the exercise, they had to search for and arrange on the board the numbers located in the central part of the square in order from the smallest to the largest while writing down only the numbers located in the lower right corner).

When taking into account the percentages of each score, similar results were reported in the case of tests conducted using the cross apparatus (tests were carried out at an imposed rate of 50 stimuli/min with the number of stimuli emitted: 49): visuomotor coordination with crossed stimuli was found to be at a low level in 12.8% of the respondents, at an average level in 19.2% of the subjects, while at a high level in as high as 69.7% of the subjects.

Tests of psychomotor reaction rime, visuomotor coordination, perceptual speed, and accuracy carried out using the Piórkowski apparatus (at an imposed rate of 107 stimuli/min in 60 s) and the cross apparatus (at an imposed rate of 50 stimuli/min with the number of stimuli emitted: 49) showed radically different results. In the case of Piórkowski’s apparatus, responses with sten scores of 1–4 were recorded in as much as 62.8% of the respondents, while sten scores of 5–6 were recorded in 37.2% of the subjects. None of the respondents scored high (7–10 sten scores). In comparison, in the case of the cross apparatus, visuomotor coordination was found to be at a low level in 12.8% of the respondents, at an average level in 19.2% of the respondents, while at a high level in as high as 69.7% of the subjects. The results are puzzling, because the correlation between the results obtained on these apparatuses (r-Pearson correlation coefficients) was found to be strong (r = 0.54).

The reaction time meter was also utilized in the study. The subjects carried out the so-called composite program no. 1, under which the correct response to the emission of a red-light signal was to press the foot button, in the case of the emission of a yellow-light signal, to press the hand button, and the emission of a green light signal and acoustic stimuli was supposed to be ignored (no response). Two parameters were recorded: average reaction time and score distribution. Based on the average reaction time, the speed, adequacy and uniformity of response to light and sound stimuli determined at a low level was found in 56.4% of the respondents, at an average level in 39.7% of the respondents, and at a high level in only 3.8% of the subjects. Based on the distribution of reaction times, the results look much better: the low level was reported in 46.2% of the subjects, while the average and high levels were found in 29.6% of the subjects each. 

A summary of the average positional measures for the studied groups of students of the Faculty of Mining, Safety Engineering, and Industrial Automation of the Silesian University of Technology is presented in Table 1.

### 4.2. The Results of Psychotechnical Tests under Time Pressure—The Study of Subjects’ Preferences Using the Analytic Hierarchy Process (AHP) Method

In parallel with the psychotechnical tests, a questionnaire survey was conducted with a random selection of respondents from among those taking part in the psychotechnical tests. The students (respondents) answered questions about the comparisons relative to each other of the “objects” (the apparatuses used in the research) and the evaluation criteria adopted (the criteria characterizing the course of the research itself) due to the difficulty occurring during the exercises. For the computational layer, the AHP method was used, under which it was assumed that the “objects” were the Piórkowski apparatus (“object” A), the cross apparatus (“object” B), and the reaction time meter (“object” C), i.e., the apparatuses used during the tests. At the same time, a set of four criteria was defined:f_1_—the complexity of the signal emitted by the apparatus;f_2_—the repeatability of the signal emitted by the apparatus;f_3_—the emission time of the signal emitted by the apparatus;f_4_—the required accuracy of the response to the signal emitted by the apparatus.

On this basis, the degree of difficulty of performing exercises on the above-mentioned “objects” (apparatuses) was determined. The results of pairwise comparisons of the evaluation criteria and “objects” (apparatuses) are presented in Table 2 and Table 3.

The assessment of compliance was made using the consistency ratio (CR) (Table 4).

Based on the results, it can be concluded that the matrix is consistent (only in the case of criterion f4, i.e., the required accuracy of response to the signal emitted by the apparatus, a value greater than 0.1 was reported), and thus, the evaluation in question can be accepted.

The final ranking of the “objects” is as follows:$$\bar{x}=\left[\begin{array}{ccc} 0.089 & 0.141 & 0.103\\ 0.324 & 0.334 & 0.681\\ 0.587 & 0.525 & 0.216 \end{array}\;\;\;\;\begin{array}{c} 0.234\\ 0.671\\ 0.094 \end{array}\right]\;\times \left[\begin{array}{c} \begin{array}{c} 0.111\\ 0.074\\ 0.593\\ 0.222 \end{array} \end{array}\right]=\left[\begin{array}{c} 0.133\\ 0.613\\ 0.253 \end{array}\right]$$

This means that, given the evaluation criteria adopted, the greatest difficulty, according to the respondents, is the implementation of the cross apparatus $\left(\bar{x}=0.613\right).\;$Exercises related to the use of the reaction time meter and the Piórkowski apparatus are significantly easier to perform—$\bar{x}$ is 0.253 for the evaluation of exercises conducted based on the reaction time meter and 0.133 for the Piórkowski apparatus.

## 5. Discussion

The article addressed the impact of time pressure—perceived as a stressor—when performing the selected psychotechnical tests. Time is an important factor taken into account when assessing visuomotor coordination, psychomotor reaction time, the ability to focus attention, and the speed and accuracy of perception (Piórkowski apparatus, cross apparatus), the speed and uniformity of reaction to light and sound stimuli (reaction time meter), or the coordination, divisibility of attention, and set-shifting (Poppelreuter tables). For the selected groups of workers, time pressure can cause psychological stress, and prolonged stress in the workplace can lead to physical and mental exhaustion of any person and then somatic complaints.

The literature review of comparable studies shows that they do not address the impact of time pressure on the conduct of psychotechnical tests.

Instead, they mainly addressed the following issues: determine the internal auditors′ dysfunctional behavior under time budget pressure, investigate how time pressure and the interaction of time pressure and nursing burnout affect patient safety, and investigate the cumulative progress of a cognitive-dynamical approach to decision making and preferential choice, called the decision field theory. The theory also accounts for the relation between choice and decision time, preference reversals between choice and certainty equivalents, and preference reversals under time pressure. The comparable studies also addressed additional issues: “does intense time pressure at work make older employees more vulnerable?”, the effects of time pressure and audit program structure on audit performance or the acquisition of survey knowledge for local and global landmark configurations under time pressure [45,46,47,48,49,50].

The research was a pilot study of a group of technical university students. After analyzing the studies that take into account the time pressure factor, it was found that none of them describe the impact on the course of psychotechnical examinations with the specific instruments and tests used in our study.

The methodology of the research was divided into two groups: first, measurements were made using the relevant diagnostic devices and tables (a sten scale was used to analyze the measurements and the distribution of results); secondly, the Analytic Hierarchy Process (AHP) method was used to study the preferences of those taking part in the psychotechnical tests. The subjects of the assessment were “objects”, i.e., the apparatuses used during the tests, and a set of specific criteria on the basis of which the degree of difficulty of performing activities using the above equipment was determined. In the process of verifying the consistency of the evaluations carried out, the CR was used.

On the basis of the research results already carried out, it can be said that the findings for the pilot group coincide with the results of the general population, i.e., people participating in this type of research (except for the assessment of coordination, attention divisibility, and attention shifting using Poppelreuter tables). It should be emphasized that the verification of the consistency of the comparisons of methods and research criteria used in the article is at a satisfactory level (it is consistent, authoritative): the values of the consistency coefficient do not exceed 0.1; the exception is the evaluation criterion No. 4, i.e., criterion f4—required accuracy of response to the signal emitted by the apparatus—for which CR = 0.194.

## 6. Conclusions

The research in question was a pilot study (it precedes the actual research), the main purpose of which was to determine how people born at the turn of the 20th and 21st centuries and entering the Faculty of Mining, Safety Engineering, and Industrial Automation at the Silesian University of Technology in Gliwice can cope with performing tasks/activities under time constraints. From the point of view of occupational health and safety and ergonomics, it seems an immensely important element of preparation for the profession, especially in the case of graduates of a technical university such as the Silesian University of Technology. Additionally, when considering infoglut—in addition to the undoubted benefits of, e.g., access to updated data and information—its quantity and intensity can have a destructive effect on human behavior (employees).

Within the article, the authors presented the results of a study assessing the predisposition of technical university students to cope with tasks under time constraints. This is due to the fact that in the future, these people may be subjected (in accordance with current legislation) to similar types of tests to assess their psychophysical fitness and ability to perform specific jobs.

The research results presented here are a prelude to the planned research on the target group (representative group). It is the intention of the authors to conduct a detailed study in the future in terms of tracking and observing the careers of graduates of the Faculty of Mining, Safety Engineering, and Industrial Automation. This information, together with data from databases containing, among other things, the research results discussed in the article, will allow for clarifying the silhouette of a graduate of the aforementioned faculty, which can be helpful, e.g., already at the stage of selecting a study major.

## Figures and Tables

**Figure 1 ijerph-19-14724-f001:**
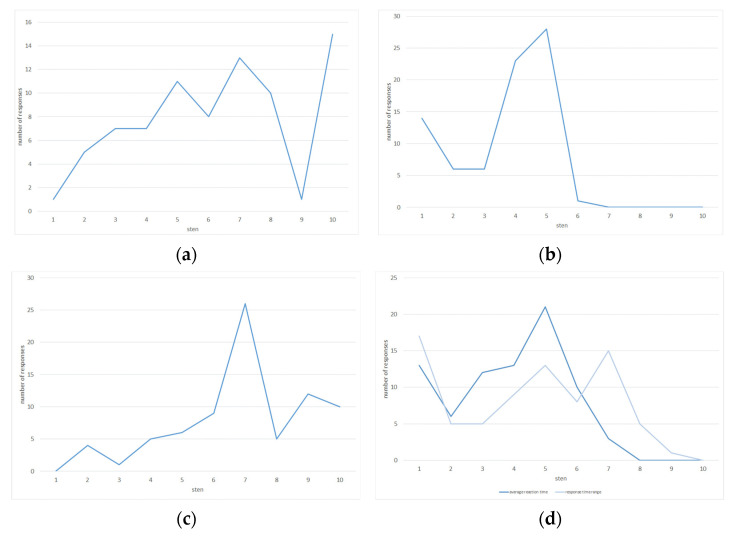
Psychotechnical tests—results: (**a**) Poppelreuter tables; (**b**) Piórkowski apparatus; (**c**) cross apparatus; (**d**) reaction time meter [own elaboration].

**Table 1 ijerph-19-14724-t001:** Average positional measures for the studied groups of students.

The Major	Parameter	Age of the Student [Age]	Poppelreuter Tables [Sten]	Piórkowski Apparatus [Sten]	Cross Apparatus [Sten]	Time Reaction Meter (Average Reaction Time) [Sten]	Time Reaction Meter (Response Time Range) [Sten]
Geodesy and Cartography	x_min_	19.00	2.00	1.00	4.00	1.00	1.00
	x_max_	25.00	10.00	5.00	10.00	6.00	8.00
	$\bar{x}$	20.48	6.86	3.43	7.10	3.52	4.14
	M(x)	20.00	7.00	4.00	7.00	4.00	5.00
	S(x)	1.22	2.40	1.56	1.72	1.82	2.73
Safety Engineering	x_min_	20.00	1.00	1.00	2.00	1.00	1.00
	x_max_	25.00	10.00	6.00	10.00	7.00	9.00
	$\bar{x}$	21.82	6.06	3.67	7.00	4.00	4.57
	M(x)	22.00	6.00	4.00	7.00	4.00	5.00
	S(x)	1.10	2.59	1.48	2.16	1.71	2.23
Mining and Geology	x_min_	22.00	3.00	1.00	2.00	1.00	1.00
	x_max_	23.00	10.00	5.00	9.00	5.00	8.00
	$\bar{x}$	22.38	6.50	3.75	6.63	3.63	4.25
	M(x)	22.00	7.00	4.50	7.00	4.50	4.50
	S(x)	0.48	2.29	1.48	2.39	1.65	2.38

where x_min_ is the minimum value, x_max_ is the maximum value, $\bar{x}$ is the arithmetic mean, M(x) is the median, and S(x) is the standard deviation.

**Table 2 ijerph-19-14724-t002:** Pairwise comparison of the importance of evaluation criteria.

	f_1_	f_2_	f_3_	f_4_
**f_1_**	1	2		
**f_2_**		1		
**f_3_**	5	5	1	6
**f_4_**	3	4		1

**Table 3 ijerph-19-14724-t003:** Pairwise comparisons of “objects” by the adopted criteria (unnormalized values).

	f_1_				f_2_				f_3_				f_4_		
	A ^1^	B ^2^	C ^3^		A	B	C		A	B	C		A	B	C
**A**	1			**A**	1			**A**	1			**A**	1		4
**B**	4	1		**B**	3	1		**B**	7	1	3	**B**	5	1	5
**C**	6	2	1	**C**	3	2	1	**C**	2		1	**C**			1

^1^ Piórkowski apparatus; ^2^. cross apparatus; ^3^ reaction time meter.

**Table 4 ijerph-19-14724-t004:** Verification of the compliance of comparisons (evaluations) carried out.

	f_1_	f_2_	f_3_	f_4_
**CR**	0.010	0.052	0.011	0.194

## Data Availability

Data supporting reported results are available on request from the study team.

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
