# Peer review of "The Impact of Time Pressure on the Results of Psychotechnical Tests Based on the Findings of Pilot Studies Conducted on a Group of Students of the Silesian University of Technology—A Case Study"

_ijerph, 2022, doi:10.3390/ijerph192214724_

Round 1
Reviewer 1 Report
Stress can be defined as a subjective and cognitive appraisal of a situation that taxes or exceeds an individual’s resources and that is experienced as overwhelming. It can result in an interplay of psychological (e.g., emotional arousal, valence) and physiological responses (e.g., variations in heartbeat, sweat production, skin temperature, and voice) which arise within seconds and may last from minutes to hours (e.g. Folkman and Lazarus, 1984, Credé et al., 2019).
It is interesting to measure this stress at work (whether during the tasks to be performed or at the workplace or afterwards). Here, we are dealing with the role of time pressure during the implementation of psychotechnical tests in a pilot phase with students, using various methods. While the overall results seem enlightening, it turns out that the approach used to achieve them requires clarification in the development of the paper, as well as in the discussion.
For example, in the introduction, the shift to psychotechnical testing is not convincingly justified. Indeed, as we know, and as the authors also point out, stress is multifactorial and can therefore come from several causes. But here the approach seems too 'quick' and does not allow for any useful consideration of both contextual and physiological factors. Moreover, the authors emphasize this further on (part 1, lines 53 and following). As a result, the work environment is mentioned, but other environments (residential, different daily activities) must also be taken into consideration.
Subsection 2.2 does not shed enough light on the context of the pilot study. What labor code is being referred to? What is its application? Why must it be considered in the protocol? Why do we have to take into account all the social and professional categories mentioned (on a ½ page) when these are elements that will not be studied later (this list should be in the appendix)? Moreover, this seems to be very context dependent, which loses in its scientific validity, especially since some profiles may appear or disappear depending on the context.
In this same sub-section, gaps are cruelly missing that would allow to establish the protocol that could then be applied. In fact, the proposed literature review does not seem to provide the reading keys to investigate the problem, nor do the hypotheses that are not included in this paper. The research protocol is thus degraded.
At the methodological level, it is specified, for example, that the authors do not take certain options as described in lines 160 and following. Why not?
The use of the AHP method is not discussed in its choice and the reader is then entitled to think that it is the only possibility of exploitation. Yet we know that multi-criteria analysis techniques are numerous (see e.g. Awasthi et al. 2018).
The results are well described and explicit. One small section, however, could be found in the methodology (lines 254-256 which finally show that the protocol was not well rendered in the previous section). In the same vein, the authors specify that they conducted a survey with a sample: but why not take all the people, and how was this random selection made?
Finally, at the level of discussion and conclusion, we are left wanting in the sense that the authors do not compare their results with comparable studies, even though the protocol should be based on certain common standards. In this sense, these parts reflect little of the work done and need to be rewritten in order to compare the results and gain in generality.
Detail: there are several typographical errors (e.g. poorly formulated sentence lines 45-46, repetitions lines 218-219).
References:
Awasthi, A., Omrani, H., Gerber, P. (2018). Multicriteria decision making for sustainability evaluation of urban mobility projects. Transportation Research Part A, 116, 247–259. https://doi.org/10.1016/J.TRA.2018.06.007
Credé, S., Thrash, T., Hölscher, C., Fabrikant, S.I., (2019). The acquisition of survey knowledge for local and global landmark configurations under time pressure. Spat. Cogn. Comput. 19, 190–219. https://doi.org/10.1080/13875868.2019.1569016
Folkman, S., Lazarus, R., 1984. Stress, appraisal, and coping. Springer Publishing Company, New York.
Author Response
Dear Reviewer,
Thank you very much for your valuable suggestions and comments on the article. We tried to make all the changes you pointed out (we have used mainly blue color in main text).
Point 1: “For example, in the introduction………must also be taken into consideration”.
Response 1:
The following response is also included in the Introduction (blue color in main text).
In addition to the mentioned factor (time pressure), there may be other causes of stress (both in daily life and at work), such as: environmental conditions, living conditions, workplace environment, work process, relationships with people, habits, and daily activities.
However, there is a group of activities where mainly time pressure has the greatest impact on the occurrence of stress, which translates into the quality of work performed. Moreover, this factor can be verified on the basis of specific and required psycho-technical tests. That is why, in accordance with current legislation, there is a need for such tests to assess the psychophysical fitness and ability of individuals to perform specific work.
Point 2: “Subsection 2.2 does not shed enough…or disappear depending on the context”.
Response 2:
The following response is also included in the subsection 2.2 (blue color in main text).
The aim of the study was to assess how students of the Faculty of Mining, Safety Engineering and Industrial Automation at the Silesian University of Technology cope with performing tasks/activities under time constraints. These individuals can become managers and executives, who will be expected, among other things, to have the right way of perceiving the environment (including divisibility of attention, ability to select information, etc.), or the ability to solve complex problem situations often under time deficit conditions. In a situation where the role of a manager/supervisor concerns mainly supervision/observation of the implemented production process and correction of its course, certain psychophysical predispositions are of particular importance. The presented research results are a prelude to the planned research on the target group (representative group). In addition, the research in question was conducted within the framework of the Project Based Learning (PBL), which significantly limited the time of conducting the research itself (participation of students). In case of continuation of the research, the authors will use additional apparatuses.
Also, some lines have been removed and replaced with the following sentence. For a detailed list of work for which special psychophysical skills are required, see the bibliographic items [22- 26].
The study refers to the Labor Code document in force in Poland, which defines the type of work that requires so-called special psychophysical fitness.
Point 3: “In this same sub-section……thus degraded”
Response 3:
The following response is also included in the subsection 2.2 (blue color in main text).
Within the framework of the article, the authors proposed to assess the predisposition of technical university students to cope with the performance of tasks under time constraints (in the first stage only as a pilot study). This is due to the fact that in the future, these people may be subjected (in accordance with the regulations in force in Poland) to a similar type of tests to assess their psychophysical fitness and ability to perform specific jobs. Moreover, the analysis of the literature indicates that studies of a similar type have not been conducted (at least studies completed with publications), which, in the opinion of the authors, allows for a conclusion that the results contained in the article are unique, and further continuation of the research – fully justified. In addition, in the Discussion and Conclusions section, references are made to the literature review of comparable studies.
Point 4: “At the methodological level……Why not?”.
Response 4:
The authors of the article intentionally gave up on some other devices used as part of standard psychotechnical research, in which reaction time plays a key role, due to the nature of the research (pilot studies). In addition, the research in question was conducted as part of the Project Based Learning (PBL), which significantly limited the time of the research itself (student participation). The authors would like to note that only the use of a cubicle darkroom is associated with adequate preparation of the subject (transition from photopic vision to scotopic vision) and necessitates each time a 13–15-minute stay of the subject (prior to the start of the study) under dark conditions (adaptation to night vision).
Point 5: “The use of the AHP method…..are numerous”.
Response 5:
The following response is also included in the subsection 3.2 (blue color in main text)
The AHP method is counted among the set of methods used in the process of solving multi-criteria decision-making problems. In this set, the following can be distinguished:
- methods using reference points, in which objects (variants) are compared with abstract reference solutions. Examples of these methods include Technique for Order Preference by Similarity to Ideal Solution (TOPSIS) and VIsekrzterijumskaOptimizacija i KompromisnoResenje (VIKOR);
- Additive methods, in which a matrix of normalized evaluations is determined, and an object (variant) is selected, for which the sum of evaluations is the highest. Examples of methods within this group are: Simple Additive Weighting Method (SAW) and Fuzzy Simple Additive Weighting Method (F-SAW);
- verbal methods based mainly on qualitative parameters, for which no objective aggregation model can be developed. The ZAPROS method (III) is an example of a method that belongs to this group;
- analytical hierarchy methods and related methods. In this group of methods, independent criteria and objects (variants) are compared with each other in pairs, which makes it possible to create a scale vector and order the objects (variants). This group includes the Analytical Hierarchy Process methods;
- methods of the ELECTRE family, in which objects (variants) are evaluated acc. to maximized criteria and the final result is a superiority relation. The final result of the method is a graph of relationships between objects;
- PROMETHEE methods. In this group of methods, objects are compared in pairs due to the adopted evaluation criteria; for each pair of objects (variants), the so-called preference flows are determined.
Point 6: “The results are well described…..selection made”.
Response 6:
Group randomization was conducted: dependent sampling (without replacement) was used.
Point 7: “Finally, at the level of discussion…. and gain in generality”.
Response 7:
We reviewed comparable studies; information is included in the Discussion (blue color in main text).
The literature review of comparable studies shows that they do not address the impact of time pressure on the conduct of psychotechnical tests.
Instead, they mainly addressed the following issues: determine the internal auditors′ dysfunctional behaviour under time budget pressure, investigated how time pressure and the interaction of time pressure and nursing burnout affect patient safety, the cumulative progress of a cognitive-dynamical approach to decision making and preferential choice called decision field theory. The theory also accounts for the relation between choice and decision time, preference reversals between choice and certainty equivalents, and preference reversals under time pressure. The comparable studies also addressed additional issues: “does intense time pressure at work make older employees more vulnerable?”, the effects of time pressure and audit program structure on audit performance or the acquisition of survey knowledge for local and global landmark configurations under time pressure.
The research was a pilot study of a group of technical university students. After analyzing the studies that take into account the time pressure factor, it was found that none of them describe the impact on the course of psychotechnical examinations with the specific instruments and tests used in our study.
Point 8: “Detail: there are several typographical errors…”.
Response 8:
Lines 45-46 (first version of the article),58-59 (second version of the article), existing sentence has been removed and replaced with another term: (blue color in main text)
H.Selye was the first to use the concept of stress in medicine in 1926, defining the term as a non-specific reaction of the body to any external requirement manifested as psychological as well as physiological reactions.
Lines 218-219 (first version of the article): repetition has been removed.
If you have any questions, please feel free to contact us. At the same time, we again ask for further evaluation and guidance.
Sincerely,
Zygmunt Korban, Eng, PhD, DSc, prof. SUoT
Maja Taraszkiewicz – Łyda, PhD

Reviewer 2 Report
It is valuable to develop technologies and methods to measure individuals’ responses to stress. So, this attempt can be meaningful. However, the research design and plan should be meticulous and the process needs to be well controlled. Therefore, in order to be published in this journal, this manuscript needs to be supplemented more. These are the things that need to be supplemented:
1. The introduction should clearly describe the necessity of this study.
2. Theoretical background is also not well presented. Readability needs to be enhanced, and it needs to be written logically. Divide paragraphs properly.
3. Unless otherwise, main text of the paper should be written in sentences and paragraphs.
4. There may be no hypothesis in the case study, but the purpose of the study was not clearly presented in this manuscript.
5. Therefore, the clinical or academic implications of the research results are not properly presented in the discussion section.
6. If you were conducting a case study, you should have detailed the subjects, that is, the characteristics of the participants.
7. Your study were not approved by the IRB, then you should have explained more specifically about conducting ethical research in detail. For example, you wrote that participants of this study were students in your school, and in that case, students should not have been forced to participate in this study.
8. In the conclusion, you don't describe anything special that you find in this study.
9. The references do not follow this journal's style properly, and there are incomplete references.
Author Response
Dear Reviewer,
Thank you very much for your valuable suggestions and comments on the article. We tried to make all the changes you pointed out (we have used mainly red color in main text).
Point 1
Response 1:
The following answer is also included in the Introduction (red color in main text).
The information relations between the object and the human in the control process, especially the perception of signals and the reactions of the worker (reaction time, the manner of response and the adequacy of response to the received signals) are the basis of occupational safety at the workstation. Human characteristics and biological effects of work are an important part of diagnosis, both in the field of work ergonomics and OSH. That is why it was so important to initially conduct the study and compile the results.
Point 2
Response 2:
The required changes to the layout have been made to improve readability.
Point 3
Response 3:
The layout of the main text has also been improved.
Point 4
Response 4:
The following response is also included in the subsection 2.2 (blue color in main text).
The aim of the study was to assess how students of the Faculty of Mining, Safety Engineering and Industrial Automation at the Silesian University of Technology cope with performing tasks/activities under time constraints. These individuals can become managers and executives, who will be expected, among other things, to have the right way of perceiving the environment (including divisibility of attention, ability to select information, etc.), or the ability to solve complex problem situations often under time deficit conditions. In a situation where the role of a manager/supervisor concerns mainly supervision/observation of the implemented production process and correction of its course, certain psychophysical predispositions are of particular importance.
Point 5
Response 5:
The following response is also included in the Discussion section (red color in main text).
On the basis of the research results already carried out, it can be said that the findings for the pilot group coincide with the results of the general population, i.e., people participating in this type of research (except for the assessment of coordination, attention divisibility and shift using Poppelreuter tables). It should be emphasized that the verification of the consistency of the comparisons of methods and research criteria used in the article is at a satisfactory level (it is consistent, authoritative): the values of the consistency coefficient do not exceed 0.1; the exception is the evaluation criterion No. 4, i.e., criterion f4 – required accuracy of response to the signal emitted by the apparatus, for which CR = 0.194.
Point 6
Response 6:
The following response (and Table 1) is also included in the “Results and research findings – a case study” section (red color in main text).
The subjects were students of technical faculties at the University of Technology. The study group was selected due to the nature of their studies (technical studies, engineering) and due to the nature of their future work. The subjects were aged 19-25 years.
A summary of the average positional measures for the studied groups of students of the Faculty of Mining, Safety Engineering and Industrial Automation of the Silesian University of Technology is presented in Table 1”Average positional measures for the studied groups of students”.
Point 7
Response 7:
Participation in the study was completely voluntary and anonymous. There were people who refused to take the test for various reasons and were not forced to do so in any way. In fact, the apparatus tests that were conducted were an interesting alternative activity for them.
Point 8
Response 8:
The following response is also included in the Conclusion section (red color in main text).
Within the article, the authors presented the results of a study assessing the predisposition of technical university students to cope with tasks under time constraints. This is due to the fact that in the future, these people may be subjected (in accordance with current legislation) to similar types of tests to assess their psychophysical fitness and ability to perform specific jobs.
The research results presented here are a prelude to the planned research on the target group (representative group). It is the intention of the authors to conduct a detailed study in the future in terms of tracking and observing the careers of graduates of the Faculty of Mining, Safety Engineering, and Industrial Automation. This information, together with data from databases containing, among other things, the research results discussed in the article, will allow for clarifying the silhouette of a graduate of the aforementioned Faculty, which can be helpful, e.g., already at the stage of selecting a study major.
Point 9
Response 9:
We have supplemented the references with further items, e.g., on similar research and related to the topics covered in this paper.
If you have any questions, please feel free to contact us. At the same time, we again ask for further evaluation and guidance.
Sincerely,
Zygmunt Korban, Eng, PhD, DSc, prof. SUoT
Maja Taraszkiewicz – Łyda, PhD

Round 2
Reviewer 2 Report
Many improvements have been made based on comments from the first round of the review.
There are still contents that seems to describe basic information that would be found in the book rather than research article, so please revise them carefully.
And, please edit carefully in accordance with the editorial policy of this journal. In particular, the citation style and reference style do not fit this journal.
Author Response
Dear Reviewer,
once more -thank you very much for your valuable suggestions and comments on the article. Please see the attachment.
Kind regards
Zygmunt Korban, Maja Taraszkiewicz-Łyda
